# The Advances of Broad-Spectrum and Hot Anti-Coronavirus Drugs

**DOI:** 10.3390/microorganisms10071294

**Published:** 2022-06-26

**Authors:** Sen Zeng, Yuwan Li, Wenhui Zhu, Zipeng Luo, Keke Wu, Xiaowen Li, Yiqi Fang, Yuwei Qin, Wenxian Chen, Zhaoyao Li, Linke Zou, Xiaodi Liu, Lin Yi, Shuangqi Fan

**Affiliations:** 1College of Veterinary Medicine, South China Agricultural University, No. 483 Wushan Road, Tianhe District, Guangzhou 510642, China; 13425951792@163.com (S.Z.); waner20191028012@stu.scau.edu.cn (Y.L.); 13970064482@163.com (W.Z.); zippo1874@163.com (Z.L.); 13660662837@163.com (K.W.); xiaowenlee@stu.scau.edu.cn (X.L.); ricky110432@163.com (Y.F.); ywqin2022@163.com (Y.Q.); chwenxian0912@163.com (W.C.); lizhaoyao@stu.scau.edu.cn (Z.L.); zlk13592585967@163.com (L.Z.); lxd18839462378@163.com (X.L.); 2Guangdong Laboratory for Lingnan Modern Agriculture, Guangzhou 510642, China; 3Key Laboratory of Zoonosis Prevention and Control of Guangdong Province, Guangzhou 510642, China

**Keywords:** antiviral drugs, coronaviruses, SARS-CoV-2, PEDV, TCM

## Abstract

Coronaviruses, mainly including severe acute respiratory syndrome virus, severe acute respiratory syndrome coronavirus 2, Middle East respiratory syndrome virus, human coronavirus OC43, chicken infectious bronchitis virus, porcine infectious gastroenteritis virus, porcine epidemic diarrhea virus, and murine hepatitis virus, can cause severe diseases in humans and livestock. The severe acute respiratory syndrome coronavirus 2 is infecting millions of human beings with high morbidity and mortality worldwide, and the multiplicity of swine epidemic diarrhea coronavirus in swine suggests that coronaviruses seriously jeopardize the safety of public health and that therapeutic intervention is urgently needed. Currently, the most effective methods of prevention and control for coronaviruses are vaccine immunization and pharmacotherapy. However, the emergence of mutated viruses reduces the effectiveness of vaccines. In addition, vaccine developments often lag behind, making it difficult to put them into use early in the outbreak. Therefore, it is meaningful to screen safe, cheap, and broad-spectrum antiviral agents for coronaviruses. This review systematically summarizes the mechanisms and state of anti-human and porcine coronavirus drugs, in order to provide theoretical support for the development of anti-coronavirus drugs and other antivirals.

## 1. Introduction

Coronaviruses were first identified in the mid-1960s and were subsequently isolated in many species, including humans, mice, swine, and chickens. Coronaviruses are capable of infecting human beings and various animals [1]. Among them, the more harmful coronaviruses include severe acute respiratory syndrome virus (SARS-CoV), severe acute respiratory syndrome coronavirus 2 (SARS-CoV-2), Middle East respiratory syndrome virus (MERS-CoV), feline infectious peritonitis virus (FIPV), and porcine coronavirus [2]. Swine coronaviruses can be divided into respiratory system coronaviruses (PRCoV) and enterovirus-type coronaviruses, such as transmissible gastroenteritis virus (TGEV), porcine epidemic diarrhea virus (PEDV), and porcine delta coronavirus (PDCoV) [3]. Patients with coronavirus infection usually present with respiratory symptoms such as fever and cough and even develop acute respiratory distress syndrome (ARDS), which leads to death [4]. TGEV, PEDV, and PDCoV share similar epidemiological, clinical, and pathological features. Similarly, their characteristic clinical symptoms are vomiting, diarrhea, and dehydration, which can cause great losses to the swine industry [5]. For these coronavirus infections, there are no specific drugs. Presently, the treatments are mostly based on vaccine immunity, in combination with drug treatment and comprehensive measures against coronavirus. However, coronaviruses are prone to antigenic variation and vaccines offer only limited cross-protection, which leads to substantial risks for the prevention and control of the diseases.

Coronaviruses belong to the genus coronavirus, family Coronaviridae, of the Nidovirales order, and can be divided into four categories, including α, β, γ, and δ, of which β-coronaviruses are further divided into A, B, C, and D four lineages [6,7]. Coronaviruses share common morphological features. They are surrounded by irregularly shaped 60–220 nm “coronals” and have 12–24 nm-long rod-like protrusions around the edges [8]. As a coated, non-segmented, positive, and single-stranded RNA virus, the length of their RNA genome can reach 30 Kb. Its genomic RNA strands contain 5′ and 3′ untranslated regions (UTRs) with a methylation “Cap” at the 5′ end, a poly tail structure at the 3′ end, and multiple open reading frames (ORFs) in the middle [9]. The ORF1a and ORF1b are responsible for the region encoding two replicates (pp1a and pp1b), occupying 2/3 of the 5′ end. The resulting polyprotein is degraded by papain-like protease (PLpro, nsp3) and 3C-like protease (3CLpro, nsp5) into 16 nonstructural proteins (nsp1-nsp16). The 3′ end mainly encodes four structural proteins, including a spike protein (S protein), a small envelope protein (E protein), a membrane protein (Mpro), and a nuclear protein (N protein) [10]. Some β coronaviruses contain an additional membrane protein, hemagglutinin esterase (HE). Dotted between these genes are genes encoding accessory proteins and the number of accessory protein genes varies among different coronaviruses.

Coronavirus S protein is composed of two subunits, the receptor-binding subunit S1 and the membrane fusion subunit S2, with S1 responsible for binding to receptors on the host cell surface, mediating attachment, and S2 responsible for viral and host cell membrane fusion, mediating invasion [11]. The S1 protein can be further divided into the N-terminal domain (NTD) and the C-terminal domain (CTD), which can bind to receptors and function as a receptor-binding domain (RBD) [12]. Recognition by the receptors’ angiotensin-converting enzyme 2 (ACE2) and dipeptidyl peptidase IV (DPP4) is associated with S1-CTD [13]. The absorptive invasion of viruses depends on RBD specificity, but different coronaviruses utilize various receptors since RBD is poorly conserved among viruses [14]. N protein can encase the viral RNA genome into a helical nucleocapsid, prolong the cell cycle in the S phase, induce endoplasmic reticulum stress, and up-regulate the expression of interleukin-8 (IL-8) and down-regulate the type I interferon [6,7,15]. Mpro is the most abundant structural protein that determines the shape of the viral envelope and is also a key protein in coronavirus assembly [15]. The E protein is the smallest structural protein in coronaviruses, which is abundantly expressed in infected cells and located in the intracellular transport site, involved in the assembly and budding of coronaviruses [16,17]. The deletion of the coronavirus E protein can significantly reduce viral production, impair viral maturation, or render progeny reproductively incompetent [18]. The four structural proteins are essential for complete virus particles and play an irreplaceable role in the viral life cycle and immune evasion, while some coronaviruses are infectious even with incomplete structural proteins [19].

The proteins of coronaviruses are not only vital in their infection process but are also targets for multiple antiviral drugs (Figure 1). Coronaviruses first bind to the host ACE2 receptor via the S protein, which undergoes cleavage fusion by the host transmembrane serine protease 2 (TMPRSS2). Then, the viruses fuse with the host cell membrane to complete their infection process [20]. In addition to direct fusion with the plasma membrane, coronaviruses invade the host via the endocytic pathway [21], which is the preferred route of SARS-CoV-2 infection [22]. The targets of anti-coronavirus drugs are divided into viral self-targets and host targets. Viral self-targets can be structural proteins of the virus or nonstructural proteins of the virus, among which the S protein and Mpro are the most studied [23]. The host targets are mainly the ACE2 receptor, TMPRSS2, cathepsin L, adaptor protein-associated kinase 1 (AAK1), cyclin G-associated kinase (GAK), phosphatidylinositol 3-phosphate 5-kinase (PIK5), 3CLpro, PLpro, RNA-dependent RNA polymerase (RdRp), and two-pore calcium channel protein 2 (TPC2) [20]. Antiviral drugs mostly exert their effects by inhibiting these targets and blocking the processes of viral invasion, replication, assembly, and release.

Currently, coronavirus mutates easily, and the vaccine development is too slow to be used in the control of emerging pandemics. Therefore, it is still of great significance to screen safe, cheap, and broad-spectrum antiviral agents against coronaviruses. This review provides a systematic overview of the mechanism and current state of anti-human and swine coronavirus drugs, including nucleoside analogs, enzyme inhibitors, antimalarial drugs, natural antioxidants, traditional Chinese medicine (TCM), and other potential antiviral agents, with a view to providing theoretical support for the development of anti-coronavirus drugs and other antiviral agents.

## 2. Nucleoside Analogs

Nucleoside analogs are synthetic, chemically modified chemotherapeutic agents that are structurally similar to endogenous purine/pyrimidine nucleosides [24]. Nucleoside analogs have been widely used to treat cancer and viral infections and the first antiviral nucleoside analogs were developed in the late 1960s [25]. Nucleoside analogs block the division of cells or the replication of viruses by impairing DNA/RNA synthesis or inhibiting cellular and viral enzymes involved in nucleoside/tide metabolism [26]. Over here, we have discussed two nucleoside analogs, remdesivir, and molnupiravir, against coronaviruses (Table 1).

### 2.1. Remdesivir

Remdesivir (Veklury^®^; Gilead Sciences, Foster City, CA, USA), a prodrug of an adenosine nucleotide analogue, is an antiviral agent with broad-spectrum activity against viruses from several families [27]. In human respiratory epithelial cells (HAE cells), the EC50 value for SARS-CoV and MERS-CoV is 74 nmol/L, and in delayed brain tumor cells, the EC50 value for mouse hepatitis virus (MHV) is 30 nmol/L [28]. The latest research shows that the remdesivir parent nucleoside effectively inhibits the proliferation of PEDV in Vero E6 cells, with EC50 of 0.31 μmol/L, and is more potent than remdesivir (EC50 = 0.74 μmol/L) [29]. As a small-molecule monophosphate prodrug, remdesivir is metabolized intracellularly to nucleoside triphosphates (the active pharmaceutical form) by the sequential reactions of ester-mediated hydrolysis and adenosine analogs, which block RdRp and nucleoside components after viruses entering into host cells, thereby achieving the goal of antiviral efficacy [30]. The antiviral activity of remdesivir against coronaviruses has rendered the drug of great interest during the current global pandemic. On 1 May 2020, based on preliminary results from phase 3 trials, the Food and Drug Administration (FDA) issued an Emergency Use Authorization (EUA) allowing the use of remdesivir for the treatment of suspected or laboratory-confirmed COVID-19 in adults and pediatric patients (over 12 years old and weighing at least 40 kg or more) hospitalized with severe disease [31]. However, on 20 November 2020, the World Health Organization stated that it is not recommended to use remdesivir for the treatment of patients with COVID-19, because there is no evidence that the drug can improve the survival rate of patients or reduce the demand for a ventilator [32]. In conclusion, remdesivir and its parent nucleoside are promising broad-spectrum antiviral nucleosides. It is necessary to modify its structure to improve its oral bioavailability in order to obtain a more efficient antiviral effect.

### 2.2. Molnupiravir

Molnupiravir, also known as EIDD-2801/MK4482, is also a ribonucleoside analog that inhibits the replication of several RNA viruses, including SARS-CoV-2 [33]. Molnupiravir was approved for marketing by the UK Medicines & Healthcare Products Regulatory Agency (MHRA) on 4 November 2021 for the treatment of adult patients with mild to moderate COVID-19 [34]. Its antiviral mechanism is the integration of an RNA base analog into the RNA genome of SARS-CoV-2, producing a defective copy of the RNA, thereby blocking the transmission of the pathogens [35]. A study has shown that molnupiravir shows significant benefits in reducing hospitalization or death in patients with mild COVID-19, whereas the effect in moderate to severe COVID-19 varies [36]. Overall, molnupiravir, the first oral antiviral agent to show significant efficacy in reducing hospitalization or death in patients with mild coronavirus disease disorder, may become an important weapon against SARS-CoV-2. However, its role in moderate to severe coronavirus disease is questionable and more research is needed, as it does not reduce the RNA content of SARS-CoV-2 and alleviate the symptoms of the patients [36].

**Table 1 microorganisms-10-01294-t001:** Antiviral nucleoside analogs and the antiviral mechanisms/targets.

Antiviral Drugs	Mechanisms/Targets	Virus	IC50/EC50 Value	Reference
Remdesivir	RdRp and nucleoside components	SARS-CoV, MERS-CoV, MHV, PEDV, SARS-CoV-2	EC50 = 0.74 μmol/L	[28,29,31]
Molnupiravir	Genomic RNA of the virus	SARS-CoV-2	-	[33,34,35]

## 3. Enzyme Inhibitors

Targeting the viral life cycle to design specific antiviral enzyme inhibitors is an important idea for antiviral drug development. Many antiviral drugs function by blocking key viral or cellular enzymes, including reverse transcriptase inhibitors, integrase inhibitors, and protease inhibitors [37]. Over here, we have discussed six enzyme inhibitors, namely Paxlovid, 3-(aminocarbonyl)-1-phenylpyridinium, 2,3-dichloroaphthoquinone, hexachlorophenol, xanthohumol, and tomatidine, against coronaviruses. Most of the enzyme inhibitors target the Mpro of coronaviruses and 3CLpro, whereas the nucleoside analog remanesivir also inhibits RdRp (Table 2).

### 3.1. Paxlovid

Paxlovid is an oral compound antiviral drug comprising nirmatrelvir and ritonavir [38]. Nirmatrelvir blocks the replication of SARS-CoV-2 and other coronaviruses by inhibiting the 3Clpro [39]. Ritonavir, an HIV-1 protease inhibitor, and also a cytochrome P450 3A4 enzyme (CYP3A4) inhibitor, is inactive against the 3CLpro and can inhibit the CYP3A4-mediated metabolism of nirmatrelvir to maintain its activity in vivo for longer periods at higher concentrations [40]. The latest clinical trial evaluation suggests that Paxlovid is currently being evaluated in phase 3 clinical trials for safety and efficacy in the treatment of non-hospitalized adult patients. Its clinical efficacy means that it is able to reduce hospitalization rates by 80% and it has also been explored as a post-exposure prophylactic agent in patients previously exposed to SARS-CoV-2 [40]. In addition, Paxlovid also demonstrated a good overall safety profile [41]. The available data suggest that Paxlovid will bring new hope for the recovery of COVID-19 patients and has the potential to be a breakthrough drug against COVID-19.

### 3.2. 3-(aminocarbonyl)-1-phenylpyridinium and 2,3-dichloroaphthoquinone

The investigators screened two compounds, 3-(aminocarbonyl)-1-phenylpyridinium and 2,3-dichloroaphthoquinone, which are capable of targeting the 3CLpro of PEDV using fluorescence resonance energy transfer (FRET)-based assays [42]. Both 3-(aminocarbonyl)-1-phenylpyridinium and 2,3-dichloroaphthoquinone exerted anti-PEDV effects by hydrogen bonding to the conserved active site (Cys144, Glu165, Gln191) of 3CLpro. Moreover, 3-(aminocarbonyl)-1-phenylpyridinium and 2,3-dichloroaphthoquinone also have an inhibitory effect on FIPV. Shi et al. also screened two new anti-PEDV compounds that were contacted by PEDV 3CLpro His41 through hydrogen bonding and hydrophobic forces and showed EC50 and CC50 values of 100 μM and 57.9 μM, respectively [42].

### 3.3. Hexachlorophenol

Hexachlorophenol is a topical anti-infective and antibacterial agent. Different concentrations of hexachlorophenol have inhibitory effects on different viruses, including the direct inactivation of rotavirus [43], the inhibition of SARS-CoV replication in Vero cells by antagonizing Mpro, and the inhibition of BK polyomavirus and simian virus 40 infections by inhibiting the ATPase activity of the large T antigen.

### 3.4. Xanthohumol

Xanthohumol, a principal prenylflavonoid isolated from hops, is an inhibitor of diacylgycerol acyltransferase (DGAT), cyclooxygenases 1 (COX1), and cyclooxygenases 2 (COX2), which possesses anti-inflammation and antioxidant activities [44]. Moreover, xanthohumol has not only antitumor and antiangiogenic effects but also antiviral activity against bovine viral diarrhea virus (BVDV), rhinovirus, HSV-1, HSV-2, and cytomegalovirus (CMV) [45,46]. With the outbreak of SARS-CoV-2, the antiviral effect of xanthohumol gained the attention of researchers. The study showed that xanthohumol restricted the replication of SARS-CoV-2 and PEDV in Vero E6 cells, with the IC50 value of 1.53 µM and 7.51 µM, respectively [47]. Xanthohumol inhibits coronaviruses by targeting and inhibiting Mpro activities. Not only that, but xanthohumol also effectively protects LPS-induced ALI against oxidative stress and inflammatory damage by activating AMPK/GSK3β, thereby inhibiting LPS-activated Txnip/NLRP3 inflammation and NF-κB signaling pathways [44]. All in all, xanthohumol is a potent pan-inhibitor of coronaviruses and an excellent lead compound for further drug development.

### 3.5. Tomatidine

Tomatidine is a steroidal alkaloid, also known as lycopene, that is extracted from the skin and leaves of tomatoes [48]. Tomatidine has various biological activities and is able to protect tomato plantations from bacteria, fungi, viruses, and certain insects during growth. Moreover, tomatidine prevents NF-κB and JNK signaling from exerting anti-inflammatory effects, and also activate autophagy in mammalian cells or Caenorhabditis Elegans [49,50]. Interestingly, tomatidine also has broad antiviral effects. Tomatidine has antiviral activities in vitro against PEDV, TGEV, porcine reproductive and respiratory syndrome virus (PRRSV), encephalomyocarditis virus (EMCV), and senecavirus A (SVA) [51]. Tomatidine’s actions against PEDV are achieved by blocking the activity of the 3CLpro through interactions with the active pocket of PEDV 3CLpro. However, tomatidine’s safety and breadth as an antiviral agent have yet to be investigated.

**Table 2 microorganisms-10-01294-t002:** Antiviral enzyme inhibitors and the antiviral mechanisms/targets.

Antiviral Drugs	Mechanisms/Targets	Virus	IC50/EC50 Value	Reference
Paxlovid	3CLpro and CYP3A4	SARS-CoV-2, HIV-1	-	[39,40]
3-(aminocarbonyl)-1-phenylpyridinium; 2,3-dichloroaphthoquinone	Mpro (Cys144, Glu165, Gln191)	PEDV, FIPV	EC50 = 100 μM	[42]
Hexachlorophenol	Mpro and ATPase	SARS-CoV	-	[43]
Xanthohumol	Mpro	BVDV, HSV-1, HSV-2, RhV, PEDV, SARS-CoV-2	IC50 = 1.53 µM and 7.51 µM	[44,45,46,47]
Tomatidine	3CLpro	PEDV, TGEV, PRRSV, EMCV, SVA, PEDV	-	[51]

## 4. Antimalarial Drugs

Antimalarial drugs, including chloroquine phosphate, primaquine phosphate, pyrimethamine, quinine, artemisinin, and artemether, are important means of controlling malaria [52]. Among them, chloroquine, hydroxychloroquine, and naphthoquine have the ability against coronaviruses. They are able to inhibit viral entry by affecting recognition by the ACE2 receptor and also influence the replication process of the viruses (Table 3).

### 4.1. Chloroquine and Hydroxychloroquine

Chloroquine (CQ) and hydroxychloroquine are antimalarial drugs that can interfere with the replication of nucleic acid, the glycosylation of viral protein, and the assembly and release of viruses, and block endosome-mediated entry by inhibiting the lysosomal weight requirement function through increasing Ph [53,54]. Increasing pH is not only detrimental to virus fusion with endosomes and entry into the cytoplasm, but also to endosomes releasing viral infectious nucleic acids and enzymes required for replication [55]. Chloroquine and hydroxychloroquine inhibit the invasion of viruses by inhibiting the binding of SARS-CoV2 to the ACE2 receptor of human cells [56]. However, chloroquine and hydroxychloroquine have an inhibitory effect that can inhibit the proliferation of T cells and reduce IFN-γ, as well as inflammatory cytokines such as TNF, IL-1, IL-6, and IL-2 [57]. They also cause gastrointestinal disorders, headaches, retinopathy, and cardiac arrhythmias among other side effects. Therefore, the FDA finally concluded that hydroxychloroquine and chloroquine are not beneficial for the treatment of COVID-19 [58].

### 4.2. Naphthoquine

Naphthoquine (NPQ), like chloroquine and hydroxychloroquine, is an antimalarial drug that was first synthesized in 1986 in a laboratory in China [59,60]. NPQ and CQ have similar chemical structures and lysosomal properties. Clinical studies have shown that the co-formulation of NPQ with artemisinin has no significant toxicity and no cardiac events and neurological events [60]. Based on the superior efficacy and safety profile of NPQ therapy, NPQ has been proposed as a new candidate for antimalarial treatment and prevention. In addition, NPQ also exhibits superior antiviral effects, especially against coronaviruses. A study showed that NPQ inhibits HCoV-229E, HCoV-OC43, and SARS-CoV-2 replication in vitro with IC50 of 2.05 ± 1.44 μM, 5.83 ± 0.74 μM, and 2.01 ± 0.38 μM, respectively [61]. NPQ has also been found to achieve a virus-inhibiting effect by affecting both the entry and post-entry replication of the virus, since it may interfere with the terminal glycosylation of ACE2 [62]. In conclusion, NPQ exhibits great advantages not only for antimalarial treatment and prevention but also for anti-coronavirus treatment. Its safety and efficacy profile may be superior to chloroquine and hydroxychloroquine, making it a superior candidate for antimalarial and anti-coronavirus drugs.

**Table 3 microorganisms-10-01294-t003:** Antiviral antimalarial drugs and the antiviral mechanisms/targets.

Antiviral Drugs	Mechanisms/Targets	Virus	IC50/EC50 Value	Reference
Chloroquine	ACE2 receptor and proinflammatory cytokines	SARS-CoV-2	-	[56]
Hydroxychloroquine				
NPQ	Affecting both entry and post-entry replication of the virus	HCoV-229E, HCoV-OC43, SARS-CoV-2	IC50 = 2.05 ± 1.44, 5.83 ± 0.74, and 2.01 ± 0.38 μM, respectively	[61]

## 5. Natural Antioxidants

Antioxidants are substances that inhibit oxidation and are also recognized as “free radical scavengers” since they form small amounts of reactive substances through free radicals [63]. Antioxidants have a variety of biological activities, such as being anti-aging, anti-tumor, anti-virus, antidiabetic, antioxidation, and immune regulation [64]. Antioxidants exert their biological functions by inhibiting the production of new free radicals (superoxide dismutase (SOD), catalase (CAT), selenium, copper, zinc), trapping free radicals to escape chain reactions (vitamins E and C, carotenoids) and restoring damages affected by free radicals (lipases, proteases) [65]. Currently, a large number of natural and synthetic antioxidants have been discovered. Here, we have discussed several natural antioxidants against coronaviruses (Table 4).

### 5.1. (−)-epigallocatechin-3-gallate, Betulinic Acid, Ursolic Acid, Aescin, Lithocholic Acid, Nordihydroguaiaretic Acid, Caffeic Acid Phenethyl Ester, and Grape Seed Extract

As a natural antioxidant, (−)-epigallocatechin-3-gallate (EGCG) is a bioactive component extracted from green tea [66]. EGCG has strong antimutagenic, antiviral, antioxidant, and anti-tumor effects and is a nontoxic and purely natural product. Studies have also found that EGCG has a significant inhibitory effect on the malignant growth and proliferation of a variety of cancer cells and can further induce tumor cells to undergo apoptosis [67,68]. In terms of being antiviral, EGCG displays potent inhibitory effects toward HIV, influenza A virus (IAV), hepatitis B virus (HBV), PRRSV, HCV, and porcine circovirus type 2 [69,70,71,72,73,74]. The latest study found that EGCG is also able to inhibit the replication of PEDV [75]. Further studies discovered that EGCG acts against PEDV infection by inhibiting PEDV attachment, entry, replication, and assembly. Interestingly, other natural antioxidants also exhibited anti-PEDV effects, including betulinic acid, ursolic acid, aescin, lithocholic acid, nordihydroguaiaretic acid, caffeic acid phenethyl ester, and grape seed extract, which were all effective in reducing the production of oxygen species induced by PEDV, which in turn inhibited the replication of PEDV in a dose-dependent manner [76].

### 5.2. Cherry Trees

Not only that, cherry trees also exhibit antiviral effects, especially red cherries. Cherry contains anthocyanins, flavane-3-alcohols, flavonols, and other phenolic compounds, as well as non-flavonoid compounds, such as hydroxycinnamic acid and hydroxybenzoic acid. Phenolics exhibit various antioxidant activities that are mainly dependent on their redox properties and are also involved in scavenging free radicals by hindering the initiation or inhibition of the propagation of lipid oxidation chain reactions [77,78]. Studies have shown that prunus yedoensis, prunus sargentii, prunus lannesiana, and prunus cerasus are able to inhibit the replication of PEDV in Vero cells by inhibiting 2,2-diphenyl-1-picrylhydrazyl (DPPH) hydroxyl radical scavenging activity, reducing power capacity and superoxide dismutase (SOD)-like activity [79].

### 5.3. Aloe Vera

Aloe vera contains a large number of anthraquinones, which have various biological properties, such as anti-inflammation, antioxidation, and immunomodulation [80,81]. Aloe vera and its extracts (quercetin, catechin hydrate, kaempferol, aloin, and emodin) have antiviral effects on a variety of viruses, including antiviral activity against IAV, pigeon paramyxovirus type 1 (PPMV-1), and herpes simplex virus type 1 (HSV-1) [82,83]. Their antiviral activities may result from the induction of antiviral genes, the inhibition of viral enzymes, and protein expression [82,83]. A recent study showed that aloe extract is able to inhibit the replication of PEDV in Vero and IPEC-J2 cells, as well as in mice, and also found that the extract not only exerted an inhibitory effect at later stages of the viral life cycle but also directly inactivated PEDV [84]. Therefore, the antiviral effect of natural antioxidants also provides suggestion for the development of drugs against SARS-CoV-2.

**Table 4 microorganisms-10-01294-t004:** Antiviral natural antioxidants and the antiviral mechanisms/targets.

Antiviral Drugs	Mechanisms/Targets	Virus	IC50/EC50 Value	Reference
EGCG	Inhibiting viral attachment, entry, replication, and assembly	HIV, IAV, HBV, HCV, PRRSV, PCV2, PEDV	-	[69,70,71,72,73,74,75]
Betulinic acid, ursolic acid, aescin, lithocholic acid, nordihydroguaiaretic acid, caffeic acid phenethyl ester, and grape seed extract	Reducing virus-induced oxygen species production	PEDV	-	[76]
Cherry trees (phenolic compounds)	Inhibiting DPPH hydroxyl radical scavenging activity, reducing power capacity, and SOD-like activity	PEDV	-	[79]
Aloe vera (anthraquinones)	Antiviral genes, viral enzymes, and proteins	IAV, PPMV-1, HSV-1, PEDV	-	[82,83,84]

## 6. Traditional Chinese Medicine (TCM)

At present, most of the anti-coronavirus drugs would cause various serious side effects or sequelae. For example, COVID-19 vaccines may cause arrhythmia, acute myocardial injury, venous thromboembolism, and other cardiovascular diseases [85]; steroid hormone therapy for SARS leads to necrosis of the femoral head [86,87]; and ribavirin has serious side effects on anemia, and so on [88]. Due to concerns about the toxicity of synthetic antiviral drugs, natural products are considered as an important source of new drugs developed to defend against viruses [89]. Since the SARS outbreak, many attempts have been made to identify treatments for coronavirus infection, among which, TCM and its extracts also take effect on anti-SARS-CoV in vitro and can effectively avoid the side effects or sequelae, which also illustrates the beneficial effects of TCM and its extracts in patients with another coronavirus [90,91]. TCM exerts antiviral effects in two aspects. On the one hand, TCM is able to exert antiviral effects directly, including directly killing viruses, blocking virus adsorption and penetration into cells, inhibiting virus replication, and blocking virus spread. On the other hand, TCM exerts antiviral effects indirectly by promoting the development of immune organs, and regulating cellular antiviral immunity and phagocytic capacity. Here, we summarize several broad-spectrum and hot TCM and their extracts targeting coronaviruses (Table 5).

### 6.1. Artemisinin and Lumefantrine

Artemisinin is the drug with the best efficacy for the treatment of malaria and has various pharmacological activities, which can also be used for the treatment of pulmonary hypertension, in addition to being antidiabetic, embryotoxic, antifungal, anti-inflammatory, anti-pulmonary fibrosis, and antibacterial, etc. [92,93]. The mechanism of artemisinin antimalarial resistance is interfering with functions such as the mitochondria surface membrane of the plasmodium. Artemisinin acts first on the food blister membrane, surface membrane, and mitochondria, and then on the nuclear envelope and endoplasmic reticulum, and it also has some effect on intranuclear chromatin, ultimately leading to the complete disintegration of parasite structure, rather than by interfering with folate metabolism in Plasmodium [93]. A recent study has found that artemisinin also plays an important role in SARS-CoV-2. Cao et al. systematically investigated the antiviral activity of several artemisinins against SARS-CoV-2 in vitro. Among them, artesunate and artemisinin B are good anti-SARS-CoV-2 drugs [94]. Interestingly, the artemisinin-related drug lumefantrine also inhibits SARS-CoV-2. Artemisinin and its analogs may exert antiviral activity by directly interfering with early protein synthesis, or by binding to host regulatory proteins to reduce early protein expression [95]. Moreover, artemisinin is also able to perturb the NF-κ B pathway [96]. For hepatitis C virus (HCV) infection, artemisinin inhibits viral replication by regulating the release of hemin and reactive oxygen species (ROS) [97,98].

### 6.2. Puerarin and Quercetin 7-Rhamnoside

Puerarin (PR), a major isoflavone isolated from the Chinese herbal medicine *Pueraria lobata*, and has antioxidant and anti-inflammatory effects [99,100,101]. PR is not only an antibacterial agent but also an antiviral candidate. For example, PR is able to protect porcine intestinal epithelial cells (IPEC-J2 cells) from enterotoxigenic *Escherichia coli* (ETEC) infection by inhibiting bacterial adhesion and inflammatory response [102], to inhibit HIV-1 replication by blocking the initial attachment of viral particles to the cell surface in primary human CD4+ T lymphocytes and macrophages [103], and to reduce in vitro hepatitis B virus (HBV) production [104]. Besides, PR also has antiviral activity against the human respiratory syncytial virus (HRSV) in human respiratory cell lines [105]. The latest study showed that PR is able to promote the proliferation of Vero cells, improve the growth performance of PEDV-infected piglets, as well as inhibit PEDV replication and IL-8 expression in vitro and in vivo, and also found that PR inhibited the activation of NF- κ B induced by PEDV [106]. An additional study showed that quercetin 7-rhamnoside (Q7R), similar to PR, is also a flavonoid that has anti-PEDV effects in vitro, and its IC50 value for anti-PEDV is 0.014 μg/mL [107]. Further studies still revealed that Q7R affects the replication of PEDV by interfering with the initial stage of PEDV infection.

### 6.3. Cepharanthine

Cepharanthine (CEP) is an anti-inflammatory and anti-tumor alkaloid that is abundant in *Stephania cepharantha* Hayata, and the security of CEP has been well verified. CEP has a broad antiviral effect spectrum and not only inhibits HIV-1 entering and binding to the central protein of heat shock protein 90 (HSP90) by reducing the fluidity of the plasma membrane, but also has a good inhibitory effect on both SARS-CoV and HCoV-OC43 [108]. Zhang et al. have found that after 48 h of coculture of CEP in Vero E6 cells, the viability of cells significantly decreased with increasing concentrations, with a TC50 value of 21.38 μg/mL [109]. It has also been shown that the EC50 value of CEP against SARS-CoV-2 in Vero E6 cells is 0.98 μM, while the EC50 of remdesivir, chloroquine, and favipiravir, which are 0.77 μM, 1.13 μM, and 61.88 μM, respectively [110]. It has been shown that CEP exerts an antiviral effect by interfering with the invasion of viruses, possibly through binding to the S protein of a virus [111,112]. On 10 May 2022, a new drug (CEP) against SARS-CoV-2 discovered by Chinese scientists was granted national invention patent authorization, and the patent inserted showed that 10 μM of CEP inhibits the replication of coronavirus by 15393-fold. In general, although the clinical efficacy has not been fully demonstrated, CEP is a promising anti-SARS-CoV-2 agent.

### 6.4. Pogostemon cablin (Blanco) Benth

*Pogostemon cablin* (Blanco) Benth. is an annual herb distributed mainly in the tropics and 62 subtropical regions of Asia and is widely cultivated in China, India, and Indonesia [113]. *Pogostemon cablin* (Blanco) Benth. is a plant of the lachnospiraceae genus spilanthes, also known as large-leaf mint, fennel, and hydro leaves, and the main medicinal part is the dried aerial [114]. Since *Pogostemon cablin* (Blanco) Benth. contains several components (hopanol, hopanone, hopanoid, and hopanoid, etc.), it similarly has various biological functions, including being anti-inflammatory, antiviral, antibacterial, and anti-tumor, and having gastrointestinal protective effects [115]. *Pogostemon cablin* (Blanco) Benth. is able to inhibit influenza virus (IV), coxsackievirus (CV), respiratory syncytial virus (RSV), HSV, and human adenoviruses (HAdVs) at a maximum effective concentration of 0.092 mg/mL in vitro with 66.76%, 86.11%, 30.56%, 19.44%, 73.25% inhibition, respectively [116]. Among the components, hopanol is a major antiviral component. The mechanism of the inhibition of virus proliferation by hopanol is to decrease the expression of RIG-I, IRF-7, and IPS-1 in the RLH pathway. Furthermore, hopanol also increases the titers of IgA, IgM, and IgG, and the levels of CD3+, CD4+, and CD8+ T cells, and suppresses the levels of TNF, IL-10, and IFN- γ in a serum against IV, reducing lung inflammation, and thereby exerting therapeutic effects against pneumonia in influenza virus-infected mice [117], which also has the effect of directly killing the virus. The latest study shows that *Pogostemon cablin* (Blanco) Benth. inhibits PEDV replication in IPEC-J2 cells and the mechanism may be the activity of enhanced antioxidant [118].

### 6.5. Cimicifuga rhizoma, Meliae cortex, Coptidis rhizoma, Phellodendron cortex, and Sophira subsprata Radix

*Cimicifuga rhizoma*, *Meliae cortex*, *Coptidis rhizoma*, *Phellodendron cortex*, and *Sophira subsprata* Radix are all traditional herbal extracts. Studies have shown that these herbal extracts significantly reduce the production of MHV and intracellular viral RNA and the expression of viral proteins in murine DBT cells with EC50 values ranging from 2.0 to 27.5 μg/mL. Furthermore, these extracts also significantly reduce the production of PEDV and vesicular stomatitis virus (VSV) production in vitro [119]. This suggests that they may contain candidate compounds for anti-coronavirus therapy.

TCM and its extracts not only have the characteristics of weak toxicity, but also possess a wide range of biological activities, and can exert antiviral effects by regulating a variety of biological processes. Therefore, there is no substitute for the potential antiviral effects of either TCM or the active ingredients. Among them, Artemisinin, PR, and CEP demonstrated the greatest anti-coronavirus potential.

**Table 5 microorganisms-10-01294-t005:** Antiviral TCM and the antiviral mechanisms/targets.

Antiviral Drugs	Mechanisms/Targets	Virus	IC50/EC50 Value	Reference
Artemisinin and lumefantrine	Early proteins	SARS-CoV-2, HCV	-	[94,95,96,97,98]
PR	-	HIV-1, HBV, HRSV, PEDV	-	[102,103,104,105,106]
Q7R	Initial stage of infection	PEDV	IC50 = 0.014 μg/mL	[107]
CEP	S protein	SARS-CoV-2, HIV-1, SARS-CoV	EC50 = 0.98 μM	[108,109,110,111,112]
*Pogostemon cablin* (Blanco) Benth.	Enhanced antioxidant activity	PEDV, IV, CV, RSV, HSV, HAdVs	-	[116,117,118]
*Cimicifuga rhizoma*, *Meliae cortex*, *Coptidis rhizoma*, *Phellodendron cortex*, and *Sophira subsprata* Radix	-	MHV, VSV, PEDV	EC50 = 2.0 to 27.5 μg/mL	[119]

## 7. Other Potential Antiviral Agents

### 7.1. Azithromycin

Azithromycin (AZM) is a second-generation broad-spectrum synthetic macrolide antibiotic that not only acts against gram-positive and gram-negative bacteria, but also has antiviral and anti-inflammatory effects and is also able to modulate immune responses [120]. AZM exerts antiviral effects by inducing type I interferon immune responses [121]. A study has shown that AZM has been widely used to treat COVID-19 patients with moderate to severe symptoms [122]. AZM is also used in combination with hydroxychloroquine, which carries a risk of malignant arrhythmias and sudden cardiac death [122]. The combination of AZM and HCQ can induce a mild to moderate reduction in heart rate, a PR interval, and a corrected QT (QTc) in vivo and vitro, and interestingly, IL-6 is more effective in reducing heart rate, increasing the PR interval and QTc [123]. Thus, in the case of IL-6 elevation during inflammation, caution must be exercised when co-administering drugs to those who known to be predisposed to arrhythmias. To reduce this risk, the antiarrhythmic drug mexiletine is combined with AZM or/and hydroxychloroquine [124]. The modality of combination therapy is positive for the antiviral effect of AZM, but there may still be some unknown toxic side effects (Table 6).

### 7.2. Losartan

Losartan, an antihypertensive drug, is able to antagonize the ACE2 receptor [125]. The envelope spike glycoprotein of SARS-CoV-2 is present on the viral surface and can bind to ACE2, thereby entering the cell [126]. Losartan, by antagonizing ACE2, is to both inhibit the infection process of SARS-CoV-2 and antagonize the metabolism of bradykinin, an inflammatory mediator, which prevents the extravasation of body fluids and leukocyte recruitment in the lung and reduces the symptoms of pulmonary edema [125,127]. Therefore, the researchers tried to use losartan as an anti-SARS-CoV-2 candidate by combining antidiabetic, anti-inflammatory, and antiviral effects to help patients recover.

### 7.3. Trichlormethiazide, D-(þ)-Biotin and Glutathione

Trichlormethiazide is an orally bioactive thiazide diuretic with antihypertensive properties. Trichlormethiazide increases the output of urine, the excretion of Na+ and K+, and improves creatinine clearance (CCRE) in rats with acute renal failure. Biotin is also known as vitamin H and vitamin B7. It is a sulfur-containing cyclic compound, and is the only D-biotin (dextranomer) existing in nature that is biologically active. Glutathione (GSH) is present in almost all mammals and is also a compound nonprotein sulfhydryl group in the body. GSH mainly has two forms, namely reduced glutathione and oxidized glutathione, the majority of which, under physiological conditions, have the functions of scavenging oxygen free radicals in the body, detoxification, and enhancing immunity. A study has shown that trichlormethiazide, D-(þ)-biotin, and GSH have an antiviral activity in PEDV-infected Vero cells, and the effective concentrations are 0.094, 0.094, and 1.5 mg/mL, respectively [128]. Furthermore, they all are able to interact with the PEDV N protein [128]. Not only that, GSH has an inhibitory effect on a variety of viruses, including Ebola virus, chikungunya virus, dengue virus, West Nile virus, SARS-CoV, and SARS-CoV-2, etc., [129,130,131,132]. The study has still shown that oral GSH not only significantly increases the levels of GSH in vivo but also enables the lymphocytes of the immune system to proliferate by about 60%, which may be the main mechanism by which it exerts antiviral effects [133].

### 7.4. Griffithsin

Griffithsin, a high mannose-specific lectin from the marine red alga Griffithsia spp, has a broad-spectrum antiviral activity that inhibits viral entry and maturation processes through binding to terminal mannoses present in high-mannose oligosaccharides and crosslinking these glycans on the surface of the viral envelope glycoproteins [129,130]. Griffithsin is currently the most potent anti-HIV drug, with a 50% EC50 in the low concentration range [134]. Griffithsin also exerts antiviral activity against multiple enveloped viruses, including SARS-CoV, MERS-COV, HCV, HSV-2, and Japanese encephalitis virus (JEV), as well as human papillomavirus [135,136,137,138,139,140,141]. Griffithsin not only shows a good safety profile, but also has no T cell activating activity. Furthermore, Griffithsin takes the advantages of little effect on the release of cytokines and chemokines from immune cells as well as excellent stability over a wide range of pH values, temperatures, and protease exposures [142,143]. These advantages make it an attractive antiviral drug. The latest study shows that Griffithsin achieves 82.8% inhibition efficiency against the infection of PEDV in Vero cells [144]. Its mechanism involves both preventing viral attachment to host cells and disrupting cell-to-cell transmission.

### 7.5. Surfactin

Surfactin, originally discovered in *Bacillus subtilis*, is a natural compound with a wide range of biological activities [145]. Surfactin is composed of heptapeptides and fatty acids in a cyclic molecule whose structure is consistent with the characteristics of wedge-shaped lipids. Two acidic amino acid residues form a relatively large hydrophilic head, while long-chain fatty acids form a relatively small hydrophobic tail [145]. This unique structural feature has a role in inhibiting membrane fusion and antiviral activity. Surfactin has been shown to inhibit a wide range of viral activities, including pseudorabies virus, porcine parvovirus, Newcastle disease virus (NDV), infectious bursal disease virus (IBDV), HSV-1, HSV-2, and TGEV [146,147]. The latest study shows that surfactin is produced in relatively low concentration ranges (15 to 50 μg/mL), inhibiting the proliferation of PEDV and TGEV in epithelial cells without cytotoxicity or disruption of the viral membrane [148]. The mechanism is to reduce the rate of viral fusion with the cell membrane and hinder the lamellar phase lipids to form negative curvatures, thereby acting as a membrane fusion inhibitor to achieve the effect of inhibiting virus proliferation. However, the potential use of surfactin as an antiviral drug is limited by its cytotoxicity. In an effort to investigate safer inhibitors of membrane fusion, investigators found that the surfactin analog SLP5, which has lower hemolytic activity, has an equivalent anti-PEDV activity and is in a 12-fold safe and effective concentration range compared to surfactin [149]. Structurally, SLP5 is a linear lipopeptide with three carboxyl groups that can obtain surfactin derivatives similar to SLP5 by hydrolysis of the lactone bonds of surfactin as well as by total synthesis. Overall, surfactin and its derivatives may be the most promising membrane fusion inhibitors identified to date, exhibiting great antiviral potential against viruses, relative to other wedge-shaped lipid membrane fusion inhibitors, such as LPC and others, which can be rapidly metabolized by all living cells to glycerophosphates or phospholipids.

### 7.6. Carbazole Alkaloids

Carbazole alkaloids are mainly distributed in coal tar and Rutaceae plants, belonging to nitrogen-containing aromatic heterocyclic compounds [150]. Carbazole alkaloids possess antibacterial, antiprotozoal, insecticidal, and antiviral activities due to their unique pharmacological activities and structural diversity [151]. Both natural and synthetic carbazole alkaloids have been reported to possess inhibitory activity against HIV, HCV, marsupial virus (CV), and HSV, etc., [152,153,154,155,156,157]. A recent study has reported that carbazole alkaloids and their analogs similarly possess antiviral activity. They were able to exert an anti-PEDV function in a dose-dependent manner in vitro, acting on the viral attachment phase of the infection of PEDV [158].

### 7.7. Exosomes

Exosomes are tiny membrane vesicles that can be secreted by most cells in the body, which have a lipid bilayer membrane, and are approximately 30–150 nmol/L in diameter. Exosomes widely exist and are distributed in various body fluids, and carry and transmit important signaling molecules. These signaling molecules form a new cell-to-cell information transmission system in all immune cells, which affects the physiological state of cells and participates in the occurrence and progression of various diseases. Exosomes are involved in many biological processes, including migration, cell growth, neuronal processes, and immune responses. It is not hard to speculate that exosomes exert functions depending on their contents. A recent study has shown that exosomes exert anti-PEDV effects through their contents, including antiviral molecules C3, C6, and CFB complexes [159]. This is a novel finding and provides new thinking for the development of drugs against SARS-CoV-2.

### 7.8. 6-Azauridine

The agent 6-azauridine is a pyrimidine analog that can inhibit a variety of viruses by inhibiting the synthesis of viral RNA. Viruses that are inhibited by 6-azauridine include another human coronavirus, HCoV-NL63 [160], FMDV by combinations of porcine interferon-alpha [161], and tick-borne flaviviruses such as the Kyasanur forest disease virus, and others [162].

### 7.9. Homoharringtonine

Homoharringtonine is an alkaloid derived from the plant *Tricuspidia tricorpus* or its congeners in the family tricorpus. Homoharringtonine has a wide range of biological activities. It not only has an inhibitory effect on the incorporation of 3H-labeled asparagine into proteins and affects the incorporation of 3H-labeled thymidine into DNA, but it also induces cell differentiation, increases the content of camp, and inhibits glycoprotein synthesis. A recent study showed that homoharringtonine is also an inhibitor against various coronaviruses, including MHV, BCoV-L9, and HECoV-4408, among others [163].

### 7.10. ZnO

ZnO is an amphoteric oxide with a high solubility acidic pH value at room temperature. Zinc is an essential trace element involved in the synthesis of more than 300 enzymes related to intestinal development and immune function [164]. In addition to zinc ions, zinc oxide molecules themselves possess antibacterial, sanitizing, and anti-inflammatory properties [165]. Interestingly, a recent study showed that ZnO administration significantly increases total superoxide dismutase activity in the ileum, and decreases the concentrations of H2O2, IL-1β, IL-6, and IL-8 in plasma, which are also capable of modulating neutrophils, thus achieving an anti-PEDV effect [166].

### 7.11. JIB-04

JIB-04 is a broad-spectrum small-molecule inhibitor of the Jumonji histone demethylase that inhibits the activity of JARID1A, JMJD2E, JMJD3, JMJD2A, JMJD2B, JMJD2C, and JMJD2D with IC50 values of 230, 340, 855, 445, 435, 1100, and 290 nM, respectively. A study found that JIB-04 is able to inhibit PEDV, TGEV, and rotavirus replication in cells, and significantly inhibits the replication of SARS-CoV-2 in Vero E6 and human bronchial epithelial cells [167]. Mechanism of action studies found that JIB-04 is able to promote the methylation of histone H3 on lysine 9 (H3K9) and lysine 27 (H3K27), initiating host antiviral responses. It was confirmed through animal experiments that either the prophylactic or therapeutic administration of JIB-04 significantly inhibited the replication of TGEV in pigs, ameliorated gastrointestinal tissue damage caused by a viral infection, and effectively alleviated the symptoms in piglets.

### 7.12. Cocktail Therapy for Coronavirus

The emergence of SARS-CoV-2 variants affecting the efficacy of the vaccine suggested the importance of developing an anti-SARS-CoV-2 therapy. Cocktail therapy is a recognized and superior home therapy and has also shown great promise in the treatment of viral infections. A recent study showed that a cocktail therapy consisting of colloidal bismuth nitrite (CBS) or bismuth subsalicylate (BSS) and *N*-acetyl-l-cysteine (NAC) is able to not only reduce the viral load and pneumonia symptoms of SARS-CoV-2 in hamster lungs but also suppress MERS-CoV, human coronavirus 229E (HCoV-229E) and SARS-CoV-2α variant (b.1.1.7) [168]. Mechanistic studies revealed that a cocktail therapy consisting of colloidal bismuth nitrite (CBS) or bismuth subsalicylate (BSS) and *N*-acetyl-l-cysteine (NAC) exhibits broad-spectrum inhibitory activity against key viral cysteinases/PLpro, Mpro, helicase (Hel), and ACE2. Cocktail therapy may play an irreplaceable role in the future treatment of COVID-19.

### 7.13. Interferon

Interferons (IFNs) are an antiviral factor produced by the host, which can be divided into type I interferons (IFN-I) and type II interferons (IFN-II) [169]. IFN-I mainly includes INF-α and INF-β, while IFN-II mainly refers to INF-γ. IFNs, which play an immunomodulatory role mainly by promoting the phagocytosis of macrophages, and have been widely used in the treatment of a variety of viral diseases, such as HBV, HCV, the herpes virus, HCoV, and so on [170,171]. IFNs exert their antiviral effects mainly by promoting the expression of antiviral proteins (2,5-oligoadenylate synthetase, protein kinases, and phosphodiesterases) through the JAK/STAT and IRF-1 signal transduction pathways. Notably, IFNs also show a significant inhibition against coronavirus. A study showed that IFN-I inhibits the infection and replication of SARS-CoV, with the intracellular viral RNA copies reduced by 50% by IFN-α at a concentration of 25 U/mL and by IFN-β at a concentration of 14 U/mL [172]. Moreover, INF-β is superior to INF-α in inhibiting the infection and replication of SARS-CoV, while MERS-CoV seems to be more sensitive to INF-α [173]. Clinically, IFN nebulization has been proved to significantly alleviate the symptoms of COVID-19, such as fever and cough, etc., [174]. Another report suggested that COVID-19 patients exhibit deficient endogenous IFN production and that most of these patients have a worse prognosis [170]. This might explain the fact that the host innate IFN response had a positive impact on the immune system for the control of COVID-19, which might be enhanced by the use of IFNs or IFN inducers. Therefore, the anti-coronavirus effects of IFNs are eagerly anticipated. Additionally, it may have better efficacy if combined with other antiviral agents.

**Table 6 microorganisms-10-01294-t006:** Potential antiviral drugs and the antiviral mechanisms/targets.

Antiviral Drugs	Mechanisms/Targets	Virus	IC50/EC50 Value	Reference
AZM	Inducing type I interferon immune responses	SARS-CoV-2	-	[120,122]
Losartan	ACE2	SARS-CoV-2	-	[125,126,127,130]
Trichlormethiazide, D- (þ) Biotin, GSH	N protein	PEDV	Its concentrations of 0.094, 0.094, and 1.5 mg/mL, respectively	[128,129,130,131,132]
Griffithsin	Preventing viral attachment to host cells and disrupting cell-to-cell transmission	HIV, SARS-CoV, MERS-COV, HCV, HSV-2, JEV, PEDV, HPV	-	[135,136,137,138,139,140,141,144]
Surfactin and SLP5	Reducing the rate of viral fusion with the cell membrane and hindering the lamellar phase lipids to form negative curvatures	PRV, PPV, NDV, IBDV, HSV-1, HSV-2, TGEV, PEDV	-	[146,147,148,149]
Carbazole alkaloids	-	HIV, HCV, CV, HSV, PEDV	-	[152,153,154,155,156,157,158]
Exosomes	C3, C6, and CFB complexes	PEDV	-	[159]
6-azauridine	Inhibiting viral RNA synthesis	HCoV-NL63, FMDV, KFDV	-	[160,161,162]
Homoharringtonine	Asparagine and thymidine	MHV, BCoV-L9, and HECoV-4408	-	[163]
ZnO	Increasing total superoxide dismutase activity	PEDV	-	[166]
JIB-04	Promoting methylation of histone H3 on lysine 9 (H3K9) and lysine 27 (H3K27), and initiating host antiviral responses	SARS-CoV-2, TGEV	-	[167]
Cocktail therapy (CBS, BBS, NAC)	PLpro, Mpro, Hel, and ACE2	SARS-CoV-2, MERS-CoV, HCoV-229E, SARS-CoV-2α Variant (b.1.1.7)	-	[168]
IFNs	Promoting the expression of antiviral proteins (2,5-oligoadenylate synthetase, protein kinases, and phosphodiesterases)	HBV, HCV, herpes virus, HCoV, SARS-CoV-2	-	[170,171,172,174]

## 8. Conclusions and Perspectives

The life cycle of coronaviruses can be divided into several phases. The early stages of the life cycle mainly consist of the virus binding to host cell receptors, endocytosis, a fusion of the viral envelope with the host cell membrane, and the release of the viral genome into the host cytoplasm. The second phase is mainly the completion of viral RNA replication, protein synthesis, and viral assembly, with the aid of the host’s energy and replicative enzymes. And then, the virus releases into the intercellular substance. According to the characteristics of the virus life cycle, we systematically summarize the broad-spectrum anti-coronavirus drugs and their mechanisms and targets, and synoptically evaluate several promising antivirals. Interestingly, cocktail therapy is a recognized and superior home therapy and has also shown great promise in the treatment of viral infections, since cocktail therapy combines the various advantages of antivirals.

Currently, there are no specific drugs available against coronaviruses, especially for SARS-CoV-2. Furthermore, coronaviruses are highly mutated, resulting in limited vaccine immune effectiveness. It poses a severe challenge to public safety and human health. Exploring broad-spectrum and hot anti-coronavirus agents and their mechanisms and targets may provide a theoretical basis for the treatment and prevention of these viral infections.

## Figures and Tables

**Figure 1 microorganisms-10-01294-f001:**
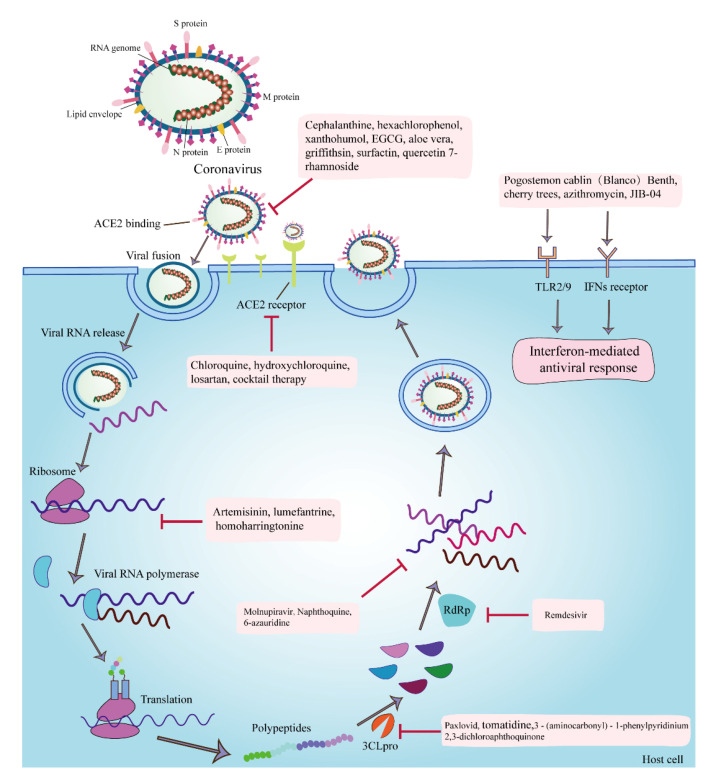
The targets of anti-coronavirus drugs and the infection processes of coronavirus. The anti-coronavirus mechanisms or targets of cepharanthine are the inhibition of viral invasion through binding to the S protein; hexachlorophenol and xanthohumol are the antagonization of the Mpro; EGCG is the inhibition of the attachment, entry, replication, and assembly of viruses; aloe vera is the direct inactivation of PEDV; Griffithsin and surfactin are the inhibition of the viral attachment; quercetin 7-rhamnoside is the inhibition of the initial stage of viral infection; chloroquine, hydroxychloroquine, losartan, and cocktail therapy for coronavirus are the inhibition of the invasion of viruses by inhibiting the binding of SARS-CoV2 to the ACE2 receptor; artemisinin, lumefantrine, and homoharringtonine are the inhibition of the synthesis of early protein; Paxlovid, tomatidine, 3-(aminocarbonyl)-1-phenylpyridinium, and 2,3-dichloroaphthoquinone are the inhibition of the 3CLpro; remdesivir is the inhibition of the RdRp and nucleoside components; molnupiravir, naphthoquine, and 6-azauridine are the inhibition of the synthesis of viral RNA; Pogostemon cablin (Blanco) Benth and JIB-04 are the stimulation of antioxidant- and H3-mediated antiviral immune responses; and azithromycin exerts antiviral effects by inducing type I interferon immune responses.

## Data Availability

Not applicable.

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
