# Peer review of "The Advances of Broad-Spectrum and Hot Anti-Coronavirus Drugs"

_microorganisms, 2022, doi:10.3390/microorganisms10071294_

Round 1
Reviewer 1 Report
review article systematizing knowledge in a selected area. A rich bibliography based on the latest reports.
In my opinion, it can be published.
Author Response
Dear reviewer :
Thank you very much for your valuable comments and your recognition of our manuscript.
Sincerely,
Sen Zeng
College of Veterinary Medicine
South China Agricultural University
Reviewer 2 Report
The author explained in detail the necessity and type of a wide range of antiviral drugs to respond to the coronavirus. It was explained in detail, so it wasn't hard to grasp the content, but I have a few questions, so I'm asking you a question.
1. In Part 2.2, Molnupiravir informed us that it was the first oral antiviral drug, while also describing its effectiveness in patients with mild coronavirus disease. Like Remdesivir in part 2.1, it would be better to understand if you describe in what ways the effect on patients with severe diseases is questionable compared to those with mild diseases.
2. Overall, the mechanisms and effects of various drugs were described. However, the explanation of the anti-infective mechanism of hexachlorophenol in Part 3.3 is insufficient, so it will be easier to understand if it is added.
3. In part 4.1. It would be good to understand if there is a supplementary explanation about the inhibition of the lysosomal weight requirement function by increasing the pH use chloroquine and hydroxychloroquine. In addition, although it is stated that Chloroquine and hydroxychloroquine inhibit the invasion of viruses by inhibiting the binding of SARS-CoV2 to the ACE2 receptor of human cells, in the back part, the validity of them is not confirmed. This part is not natural in the context. It would be better the flow of the sentences could change more naturally.
4. In part 4.2, Just as it explains the mechanism by which chloroquine and hydroxychloroquine interact with the virus, it will be easier for the reader to understand if we also describe the mechanism by which NPQ inhibits viral replication.
5. In part 5, I'm curious about the criteria for selecting the following three natural antioxidants: (-)-Epigallocatechin-3-gallate, betulinic acid, ursolic acid, aescin, lithocholic acid, 292 nordihydroguaiaretic acid, caffeic acid phenethyl ester, and grape seed extract, Cherry trees and Aloe Vera. In addition, it was explained that it has an antiviral effect by inhibiting the replication of PEDV in common, and I wonder how it inhibits the replication and whether it does not show an antiviral effect in other types of viruses other than PEDV.
6. In part 6, If you explain whether TCM's weak toxicity is relative to the toxicity of synthetic antiviral drugs or harmless to the human body when used as a drug, it will help us understand.
7. Part 7.1 said that Azithromycin is used together with hydroxychloroquine, but I wish there was an explanation as to why it is used together with drugs with the risk of malignant arrhythmias or sudden cardiac death.
8. In part 7.4, Griffithsin has antiviral activity by inhibiting HIV entry. I am curious about the additional antiviral mechanism for this. I also want to know additional information about Griffithsin particularly having broad-spectrum antiviral activity

Author Response
Dear reviewer,
Thank you very much for your valuable comments on our manuscript. We have made relevant changes one by one based on your suggestions. You can view the details in the PDF attachment.
Please see the attachment.
We appreciate your warm work earnestly and hope that the correction will meet with approval.
Once again, thank you very much for your comments and suggestions.
Sincerely,
Sen Zeng
College of Veterinary Medicine
South China Agricultural University

Reviewer 3 Report
The review is very interesting. In future it could be improved by adding some information on immunotherapetic approaches. For instance, use of interferon and other immunomodulatory compounds.
Author Response
Dear reviewer,
Thank you very much for your valuable suggestions on our manuscript. We have made relevant changes based on your suggestions. The details are as follows.
- The review is very interesting. In future it could be improved by adding some information on immunotherapetic approaches. For instance, use of interferon and other immunomodulatory compounds
We have added the information on interferons against coronavirus that “Interferons (IFNs) are an antiviral factor produced by the host, which can be divided into type I interferon (IFN-I) and type II interferon (IFN-II) [169]. IFN-I mainly includes INF-α and INF-β, while IFN-II mainly refers to INF-γ. IFNs, which play an immunomodulatory role mainly by promoting phagocytosis of macrophages, have been widely used in the treatment of a variety of viral diseases, such as HBV, HCV, herpes virus, HCoV, and so on [170,171]. IFNs exert their antiviral effects mainly by promoting the expression of antiviral proteins (2,5-oligoadenylate synthetase, protein kinases, and phosphodiesterases) through the JAK / STAT and IRF-1 signal transduction pathways. Notably, IFNs also show a significant inhibition against coronavirus. A study showed that IFN-I inhibits the infection and replication of SARS-CoV with the intracellular viral RNA copies are reduced 50% by IFN-α at a concentration of 25 U/ml and by IFN-β at a concentration of 14 U/ml [172]. Moreover, INF-β is superior to INF-α in inhibiting the infection and replication of SARS-CoV, while MERS-CoV seems to be more sensitive to INF-α [173]. Clinically, IFN nebulization has been proved to significantly alleviate the symptoms of COVID-19, such as fever, cough, etc [174]. Another report suggested that COVID-19 patients exhibit deficient endogenous IFN production and that most of these patients have a worse prognosis [170]. This might explain the fact that the host innate IFN response had a positive impact on the immune system for the control of COVID-19, which might be enhanced by the use of IFNs or IFN inducers. Therefore, the anti-coronavirus effects of IFNs are eagerly anticipated. And it may have better efficacy if combined with other anti-viral agents.”
We believed that our manuscript has been significantly improved.
We appreciate your warm work earnestly and hope that the correction will meet with approval.
Once again, thank you very much for your comments and suggestions.
Sincerely,
Sen Zeng
College of Veterinary Medicine
South China Agricultural University